# Long-term survival of female versus male patients after coronary artery bypass grafting

**Armando Abreu**[1,2]*, **José Máximo**[1,2]☯, **Adelino Leite-Moreira**[1,2]☯

**1** Department of Surgery and Physiology, Faculty of Medicine of the University of Porto, Porto, Portugal,
**2** Department of Cardiothoracic Surgery, Centro Hospitalar Universitário S. João, Porto, Portugal

☯ These authors contributed equally to this work.
* armandopabreu@gmail.com

**Data Availability Statement:** All data are contained in the paper and/or Supporting Information files.

**Funding:** The authors received no specific funding for this work.

## Abstract

### Background

Several of the most extensively used risk prediction tools for coronary artery bypass grafting outcomes include female sex as an independent risk factor for postoperative outcomes. It is not clear whether this putative increased surgical risk impacts long-term survival. This study aimed to assess sex differences in 10-year all-cause mortality.

### Methods

Retrospective analysis of 5340 consecutive patients undergoing primary isolated coronary artery bypass surgery, performed from 2000 to 2015, in a Portuguese level III Hospital. The primary endpoint was all-cause mortality at ten years. We employed an overlap weighting algorithm to minimize confounding. Its target population highlights patients with the most overlap in their observed characteristics, and its corresponding estimand is the average treatment effect in the overlap population.

### Results

We identified that 5340 patients underwent isolated CABG: 1104 (20.7%) were female, and 4236 (79.3%) were male. Sixteen patients were lost to follow-up (0.3%). The median follow-up time was 12.79 (IQR, 9.52–16.66) years: 12.68 (IQR, 9.48–16.54) years for the male patient group and 13.13 (IQR, 9.75–16.98) years for the female patient group. The primary endpoint of all-cause mortality at ten years occurred in 1106 patients (26.1%) in the male patient group, compared with 315 (28.5%) in the female patient group. The unweighted survival analysis for both groups reveals the worst long-term prognosis for the female cohort (hazard ratio, 1.22; 95% CI, 1.10 to 1.35; p < 0.001), while in the overlap weighted survival analysis, such long-term difference in prognosis disappears (hazard ratio, 0.98; 95% CI, 0.88 to 1.09; p = 0.693).

### Conclusion

In this longitudinal, population-level analysis of patients undergoing primary, isolated CABG, we demonstrated that the female sex is not associated with increased long-term all-

**Competing interests:** The authors have declared that no competing interests exist.

cause mortality compared to their male counterparts. Thus, sex should not influence the undertaking of an adequate revascularization strategy.

## Introduction

A consistent finding among patients undergoing coronary artery bypass grafting (CABG) is the superior perioperative mortality seen in female patients compared to male patients [1–3]. Women are likely to present at an older age with a more significant burden of cardiovascular comorbidities and a worse functional status at baseline [4]. Consequently, two of the most extensively used risk prediction tools for CABG outcomes, the Society of Thoracic Surgeons (STS) score and the EuroSCORE, include female sex as an independent risk factor for post-CABG outcomes [5–7]. It is not clear whether this putative increased surgical risk impacts long-term survival.

Multiple studies reported the impact of the female gender on long-term clinical outcomes after isolated CABG with conflicting results. A meta-analysis of 20 observational studies documented that women who underwent isolated CABG experienced higher mortality at short-term and long-term follow-up than their male counterparts [8]; on the other hand, other more recent studies report that the female sex does not constitute a significant predictor of long-term prognosis [9–11]. Therefore, the role of sex on long-term clinical outcomes after CABG remains uncertain and deserves further clarification.

This study compares 10-year survival in female and male patients with ischemic heart disease admitted to primary coronary artery bypass grafting, in a single level III institution, between 2000 and 2015. We employed an overlap weighting (OW) algorithm to minimize confounding. Its target population highlights patients with the most overlap in their observed characteristics, and its corresponding estimand is the average treatment effect (ATE) in the overlap population [12].

## Methods

### Ethics

Our Institution's Ethics Committee approved this research, and the need for informed consent was waived.

### Study design

We conducted an observational retrospective study to evaluate sex-related differences in baseline characteristics, utilization trends, in-hospital complications, length of hospital stay, discharge disposition, and long-term (10 years) survival in patients with isolated coronary artery disease undergoing CABG. Thus, we analyzed an administrative dataset containing all hospitalizations occurring in a level III hospital from January 1, 2000, to September 30, 2015 (chosen as the cutoff date because of ICD-10-CM implementation). The corresponding diagnoses and procedures were coded for each hospitalization based on the International Classification of Diseases, 9th Revision, Clinical Modification (ICD-9-CM).

### Study population

Patients were included in the study if they underwent primary coronary artery bypass surgery (*International Classification of Diseases*, *Ninth Revision*, *Clinical Modification* [ICD-9-CM]

codes 36.10, 36.11, 36.12, 36.13, 36.14, 36.15, 36.16, 36.17 or 36.19) during the study period. Exclusion criteria included previous cardiac surgery, concomitant valve replacement or repair, concurrent aorta surgery, and simultaneous correction of myocardial infarction mechanical complications (S1 Table).

## Data sources and variables

From an administrative dataset containing all hospitalizations occurring in our Institution from 2000 to 2015, we identified all hospitalizations with at least one associated procedure code of CABG. The predictive or independent variable was the sex of the patient. We obtained patients' baseline characteristics from our institution patient's discharge datasets. After extracting the relevant ICD-9-CM codes, we computed the Charlson Comorbidity Index using the Quan et al. coding scheme [13, 14]. We provide definitions of coexisting conditions in S2 Table.

## Outcomes

We compared episodes concerning female patients to those of male patients. The primary outcome variable was 10-year survival. The patient discharge database was linked to the National Patient Registry (RNU) to ascertain patient life status. Secondary outcomes included a set of predefined in-hospital complications (see S3 Table for detailed definitions), the length of hospital stay, and discharge disposition (categorized as home discharge, transfer to other healthcare facilities or in-hospital death).

## Statistical analysis

Data are presented as absolute frequencies and percentages for categorical variables and as means and standard deviations or median and interquartile range, where appropriate, for continuous variables. We used the standardized mean difference to assess discrepancies in covariates between treatment groups, as it allows for the judgment of the relative balance of variables measured in different units. We held values less than 0.1 to indicate a negligible difference in the mean or frequency of a covariate between treatment groups [15].

In the univariate analysis, we computed summary measures of risk (odds ratio), and its associated 95% confidence interval, using simple logistic regression for each predefined outcome.

We performed overlap propensity score weighting to address potential confounding. The properties of overlap weights relative to inverse probability weighting include improved covariate balance and increased precision of effect measures estimates [12]. Multivariable logistic regression was used in each treatment group to estimate each patient's propensity score. The propensity model included the following variables: age, admission status (scheduled vs unscheduled), disease presentation (stable coronary disease, unstable angina / NSTEMI, and STEMI), hypertension, diabetes mellitus, hyperlipidemia, obesity, smoking history, cerebrovascular disease, congestive heart failure, chronic obstructive pulmonary disease, peripheral vascular disease, chronic kidney disease, liver disease, anemia, coagulation disorders, cancer history, the Charlson comorbidity index, use of cardiopulmonary bypass (i.e., whether the procedure was performed off-pump or on-pump), number of internal mammary arteries used (i.e., none, single, or bilateral), and the total number of grafts performed. Finally, we assessed the balance between treatment groups using standardized mean differences, with an ideal balance represented by a standardized difference of 10% or less. We included visual depictions of distributional balance as they are a helpful complement to numerical summaries [16].

We derived weighted logistic regression models with a robust variance estimator with the outcome as the dependent variable and the group on which the propensity score balances (e.g., the treatment group) as the only independent/predictor variable [17–19].

Estimates of survival probabilities were calculated using the Kaplan–Meier method and compared with the log-rank test [20–22]. Follow-up time, described by median and interquartile range, was obtained using the same estimator by reversing the event indicator so that the outcome of interest became being censored [23]. We employed a weighted Cox proportional hazards regression model with a robust variance estimator to compare long-term mortality between groups [17–19].

*P* values were two-sided with a significance threshold of 0.05. All statistical analyses were performed using R version 4.1.3 [24].

## Results

Between 2000 and 2015, 5340 patients underwent isolated CABG: 1104 (20.7%) were female, and 4236 (79.3%) were male patients (Fig 1). This relative difference persisted during the study period (Fig 2).

### Baseline characteristics

Regarding baseline characteristics (Table 1), women were older (66.7 ± 9.4 vs 62.9 ± 10.0), were more likely to have hypertension (77.4% vs 64.5%; SMD 0.287), diabetes mellitus (50.4% vs 35.5%; SMD 0.317), excessive body weight (30.1% vs 20.9%; SMD 0.212), and anemia

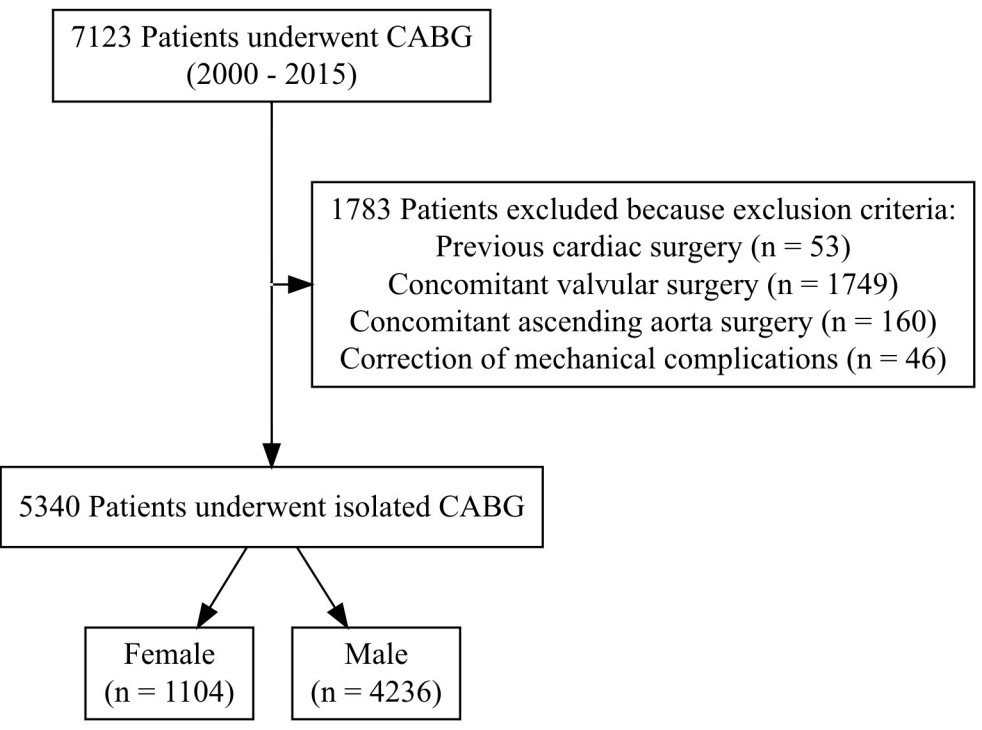

**Fig 1. Study flowchart.** Exclusion criteria for 7123 patients who underwent coronary artery bypass grafting (CABG) in Northern Portugal.

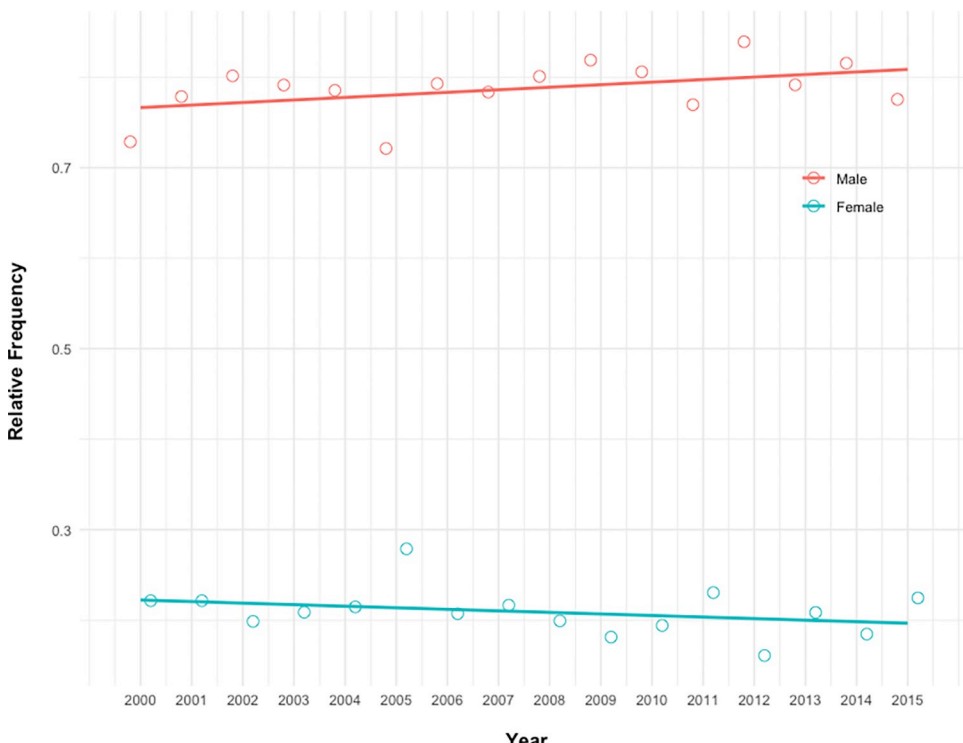

**Fig 2. Temporal trends.** Temporal trends in the relative frequency of isolated coronary artery grafting (CABG) by sex.

**Table 1. Baseline characteristics.**

| | Unadjusted | | | Overlap weighted | | |
|---|---|---|---|---|---|---|
| | **Male** | **Female** | | **Male** | **Female** | |
| **Characteristic** | **(n = 4236)** | **(n = 1104)** | **SMD[k]** | **(n = 726.9)** | **(n = 726.9)** | **SMD[k]** |
| **Age**, mean (SD[a]) | 62.9 (10.0) | 66.7 (9.4) | 0.391 | 66.0 (9.1) | 66.0 (9.8) | 0 |
| **Admission**, n (%) | | | 0.046 | | | 0 |
| Scheduled | 1988 (46.9) | 493 (44.7) | | 407.5 (46.4) | 407.5 (46.4) | |
| Unscheduled | 2248 (53.1) | 611 (55.3) | | 389.4 (53.6) | 389.4 (53.6) | |
| **Presentation**, n (%) | | | 0.042 | | | 0 |
| Chronic CAD[b] | 2705 (63.9) | 702 (63.6) | | 477.2 (65.7) | 469.7 (64.6) | |
| UA / NSTEMI[c] | 1112 (26.3) | 304 (27.5) | | 180.1 (24.8) | 195.2 (26.9) | |
| STEMI[d] | 419 (9.9) | 98 (8.9) | | 69.5 (9.6) | 62.0 (8.5) | |
| **Hypertension**, n (%) | 2731 (64.5) | 854 (77.4) | 0.287 | 537.8 (74.0) | 537.8 (74.0) | 0 |
| **Diabetes mellitus**, n (%) | | | 0.317 | | | 0 |
| No diabetes | 2733 (64.5) | 547 (49.5) | | 394.9 (54.3) | 391.8 (53.9) | |
| Non-insulin-treated | 1322 (31.2) | 465 (42.1) | | 282.0 (38.8) | 288.1 (39.6) | |
| Insulin-treated | 181 (4.3) | 92 (8.3) | | 50.0 (6.9) | 47.0 (6.5) | |
| **Hyperlipidemia**, n (%) | 2645 (62.4) | 736 (66.7) | 0.088 | 465.4 (64.0) | 465.4 (64.0) | 0 |
| **Obesity**, n (%) | 885 (20.9) | 332 (30.1) | 0.212 | 190.8 (26.3) | 190.8 (26.3) | 0 |
| **Smoking history**, n (%) | | | 1.127 | | | 0 |
| No smoking history | 2181 (51.5) | 1047 (94.8) | | 642.0 (88.3) | 671.6 (92.4) | |
| Previous smoker | 1163 (27.5) | 19 (1.7) | | 76.9 (10.6) | 17.7 (2.4) | |
| Current smoker | 892 (21.1) | 38 (3.4) | | 8.0 (1.1) | 37.6 (5.2) | |
| **CVD[e]**, n (%) | 688 (16.2) | 185 (16.8) | 0.014 | 119.6 (16.4) | 119.6 (16.4) | 0 |
| **CHF[f]**, n (%) | 832 (19.6) | 211 (19.1) | 0.013 | 139.1 (19.1) | 139.1 (19.1) | 0 |
| **COPD[g]**, n (%) | 323 (7.6) | 60 (5.4) | 0.089 | 41.0 (5.6) | 41.0 (5.6) | 0 |

*(Continued)*

**Table 1.** (Continued)

| | Unadjusted | | | Overlap weighted | | |
|---|---|---|---|---|---|---|
| | **Male** | **Female** | | **Male** | **Female** | |
| **Characteristic** | **(n = 4236)** | **(n = 1104)** | **SMD[k]** | **(n = 726.9)** | **(n = 726.9)** | **SMD[k]** |
| **PVD[h]**, n (%) | 187 (4.4) | 34 (3.1) | 0.070 | 22.4 (3.1) | 22.4 (3.1) | 0 |
| **CKD[i]**, n (%) | | | 0.069 | | | 0 |
| No chronic kidney disease | 4010 (94.7) | 1033 (93.6) | | 681.9 (93.8) | 680.4 (93.6) | |
| Non-dialysis dependent | 196 (4.6) | 66 (6.0) | | 40.2 (5.5) | 43.2 (5.9) | |
| Dialysis dependent | 30 (0.7) | 5 (0.5) | | 4.8 (0.7) | 3.3 (0.5) | |
| **Liver disease**, n (%) | 89 (2.1) | 25 (2.3) | 0.011 | 15.3 (2.1) | 15.3 (2.1) | 0 |
| **Anemia**, n (%) | 363 (8.6) | 135 (12.2) | 0.120 | 78.2 (10.8) | 78.2 (10.8) | 0 |
| **Coagulation disorders**, n (%) | 68 (1.6) | 25 (2.3) | 0.048 | 14.1 (1.9) | 14.1 (1.9) | 0 |
| **Cancer**, n (%) | 46 (1.1) | 7 (0.6) | 0.049 | 5.6 (0.8) | 5.6 (0.8) | 0 |
| **CCI[j]**, mean (SD[a]) | 4.14 (1.70) | 4.57 (1.57) | 0.266 | 4.46 (1.63) | 4.46 (1.59) | 0 |

Baseline characteristics for unweighted and overlap weighted cohorts: male vs female.

[a] standard deviation

[b] coronary artery disease

[c] unstable angina/non-ST-elevation myocardial infarction

[d] ST-elevation myocardial infarction

[e] cerebrovascular disease

[f] congestive heart disease

[g] chronic obstructive pulmonary disease

[h] peripheral vascular disease

[i] chronic kidney disease

[j] Charlson comorbidity index

[k] standardized mean difference

**Table 2. Procedural details.**

| | Unadjusted | | | Overlap weighted | | |
|---|---|---|---|---|---|---|
| | *Male* | *Female* | | *Male* | *Female* | |
| **Characteristic** | *(n = 4236)* | *(n = 1104)* | **SMD[h]** | *(n = 726.9)* | *(n = 726.9)* | *SMD* |
| **CBP[a]**, n (%) | | | 0.003 | | | 0 |
| OPCAB[b] | 1575 (37.2) | 409 (37.0) | | 266.7 (36.7) | 266.7 (36.7) | |
| ONCAB[c] | 2661 (62.8) | 695 (63.0) | | 460.2 (63.3) | 460.2 (63.3) | |
| **IMA utilization**, n (%) | | | 0.251 | | | 0 |
| No IMA[d] | 94 (2.2) | 51 (4.6) | | 22.1 (3.0) | 29.5 (4.1) | |
| SIMA[e] | 2932 (69.2) | 843 (76.4) | | 560.5 (77.1) | 545.6 (75.1) | |
| BIMA[f] | 1210 (28.6) | 210 (19.0) | | 144.4 (19.9) | 151.8 (20.9) | |
| **Distal anastomosis**, mean (SD[g]) | 2.55 (0.86) | 2.40 (0.85) | 0.179 | 2.45 (0.84) | 2.45 (0.86) | 0 |

Procedural details for unweighted and overlap weighted cohorts: male vs female.

[a] cardiopulmonary bypass

[b] off-pump coronary artery bypass

[c] on-pump coronary artery bypass

[d] internal mammary artery

[e] single internal mammary artery

[f] bilateral internal mammary artery

[g] standard deviation

[h] standardized mean difference

(12.2% vs 8.6%; SMD 0.120). This disproportionate risk profile translated into a higher Charlson comorbidity index (4.57 ± 1.57 vs 4.14 ± 1.70; SMD 0.266).

Considering intra-operative procedure details (Table 2), although there was no significant difference in the relative utilization of OPCAB or ONCAB techniques between groups, there was a higher proportion of women not receiving any internal mammary artery graft (4.6% vs 2.2%), and a lower proportion of women received bilateral internal artery mammary grafting (19.0% vs 28.6%; SMD 0.251). Similarly, the mean number of grafts performed was lower in the female cohort (2.40 ± 0.85 vs 2.55 ± 0.86; SMD 0.179).

## Crude outcome analysis

In the crude outcome analysis (Table 3), women had 25% higher odds of requiring a blood transfusion (30.8% vs 26.3%; OR = 1.25; 95% CI 1.08, 1.44; p = 0.003) and 74% higher odds of having a surgical wound complication (2.4% vs 1.4%; OR = 1.74; 95% CI 1.07, 2.74; p = 0.021) in the index hospitalization. Likewise, women had higher odds of being discharged to another healthcare facility following the index hospitalization (7.4% vs 3.7%; OR 2.08; 95% CI 1.58,

**Table 3. Crude outcome analysis.**

| Outcome | Male (n = 4236) | Female (n = 1104) | OR[f] | (95% CI[h]) | p |
|---|---|---|---|---|---|
| Stroke, n (%) | 44 (1.0) | 6 (0.5) | 0.52 | 0.20, 1.13 | 0.135 |
| Cardiac, n (%) | | | | | |
| POAF[a] | 528 (12.5) | 151 (13.7) | 1.11 | 0.91, 1.35 | 0.282 |
| Pacemaker implantation | 17 (0.4) | 8 (0.7) | 1.81 | 0.74, 4.08 | 0.167 |
| IABP[b] | 137 (3.2) | 40 (3.6) | 1.12 | 0.78, 1.59 | 0.520 |
| Cardiac arrest | 19 (0.4) | 3 (0.3) | 0.60 | 0.14, 1.78 | 0.419 |
| Respiratory, n (%) | | | | | |
| Prolonged ventilation | 184 (4.3) | 47 (4.3) | 0.98 | 0.70, 1.35 | 0.900 |
| Reintubation | 101 (2.4) | 21 (1.9) | 0.79 | 0.48, 1.25 | 0.341 |
| Tracheotomy | 21 (0.5) | 4 (0.4) | 0.73 | 0.21, 1.92 | 0.565 |
| Acute kidney injury, n (%) | 37 (0.9) | 12 (1.1) | 1.25 | 0.62, 2.33 | 0.508 |
| Hemorrhage, n (%) | 158 (3.7) | 35 (3.2) | 0.85 | 0.57, 1.21 | 0.375 |
| RBC[c] transfusion, n (%) | 1114 (26.3) | 340 (30.8) | 1.25 | 1.08, 1.44 | 0.003 |
| Surgical wound, n (%) | 58 (1.4) | 26 (2.4) | 1.74 | 1.07, 2.74 | 0.021 |
| Discharge disposition, n (%) | | | | | < 0.001 |
| Home | 4013 (94.7) | 1002 (90.8) | 0.55 | 0.43, 0.70 | |
| Other hospital | 157 (3.7) | 82 (7.4) | 2.08 | 1.58, 2.74 | |
| Death | 66 (1.6) | 20 (1.8) | 1.17 | 0.69, 1.89 | |
| Outcome | Male (n = 4236) | Female (n = 1104) | CIE[g] | (95% CI[h]) | p |
| LOS[d], median (IQR[e]) | 7 (6, 9) | 7 (6, 10) | 1.17 | 0.57, 1.78 | < 0.001 |

Crude outcome analysis (unweighted cohort): male vs female.

[a] postoperative atrial fibrillation

[b] intra-aortic balloon pump counterpulsation

[c] red blood cell

[d] length of stay

[e] interquartile range

[f] odds ratio

[g] change in estimate

[h] confidence interval

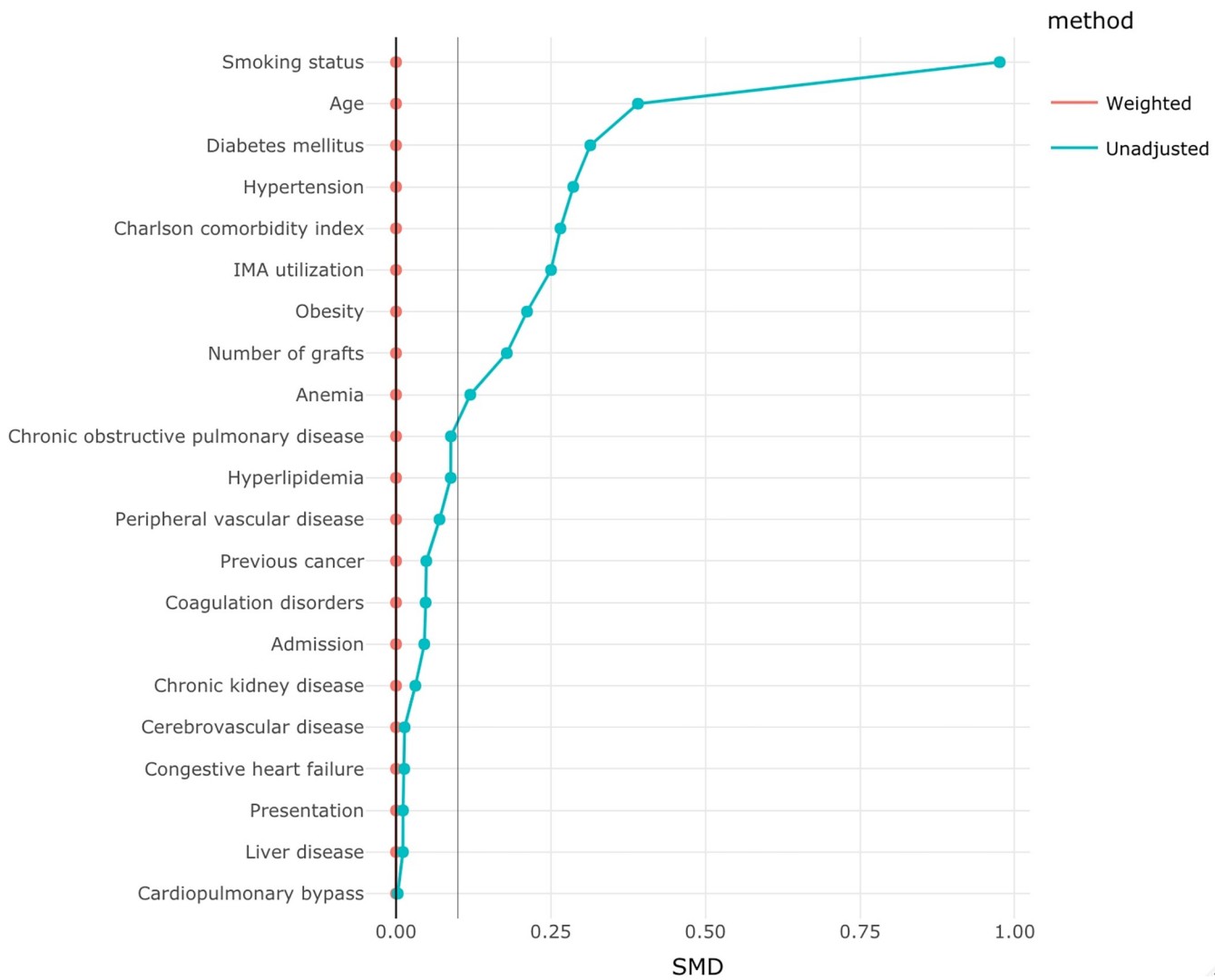

**Fig 3. Covariate balance.** Covariate balance: unweighted vs overlap weighted cohorts.

2.74; p < 0.001). On the other hand, women required longer hospitalization periods [7 (IQR 6, 10) days vs 7 (6, 9) days; CIE 1.17 days; 95% CI 0.57, 1.78; p < 0.001].

## Weighted outcome analysis

Overlap weighting balanced baseline characteristics in each group (Table 1 and Fig 3). As depicted in Fig 4, female patients had 18% more odds of requiring an RBC transfusion (OR 1.18, 95% CI 1.00, 1.38, p = 0.046), and 73% more odds of having surgical wound complications (OR 1.73, 95% CI 1.02, 2.92, p = 0.042). We have not noted any other differences in pre-specified complications or discharge disposition rates. Concerning the length of hospital stay, there were no significant differences between groups (CIE 0.59 days, 95% CI -0.12–1.31, p = 0.105).

## Survival analysis

Sixteen patients were lost to follow-up (0.3%). The median follow-up time was 12.79 (IQR, 9.52–16.66) years: 12.68 (IQR, 9.48–16.54) years for the male patient group and 13.13 (IQR,

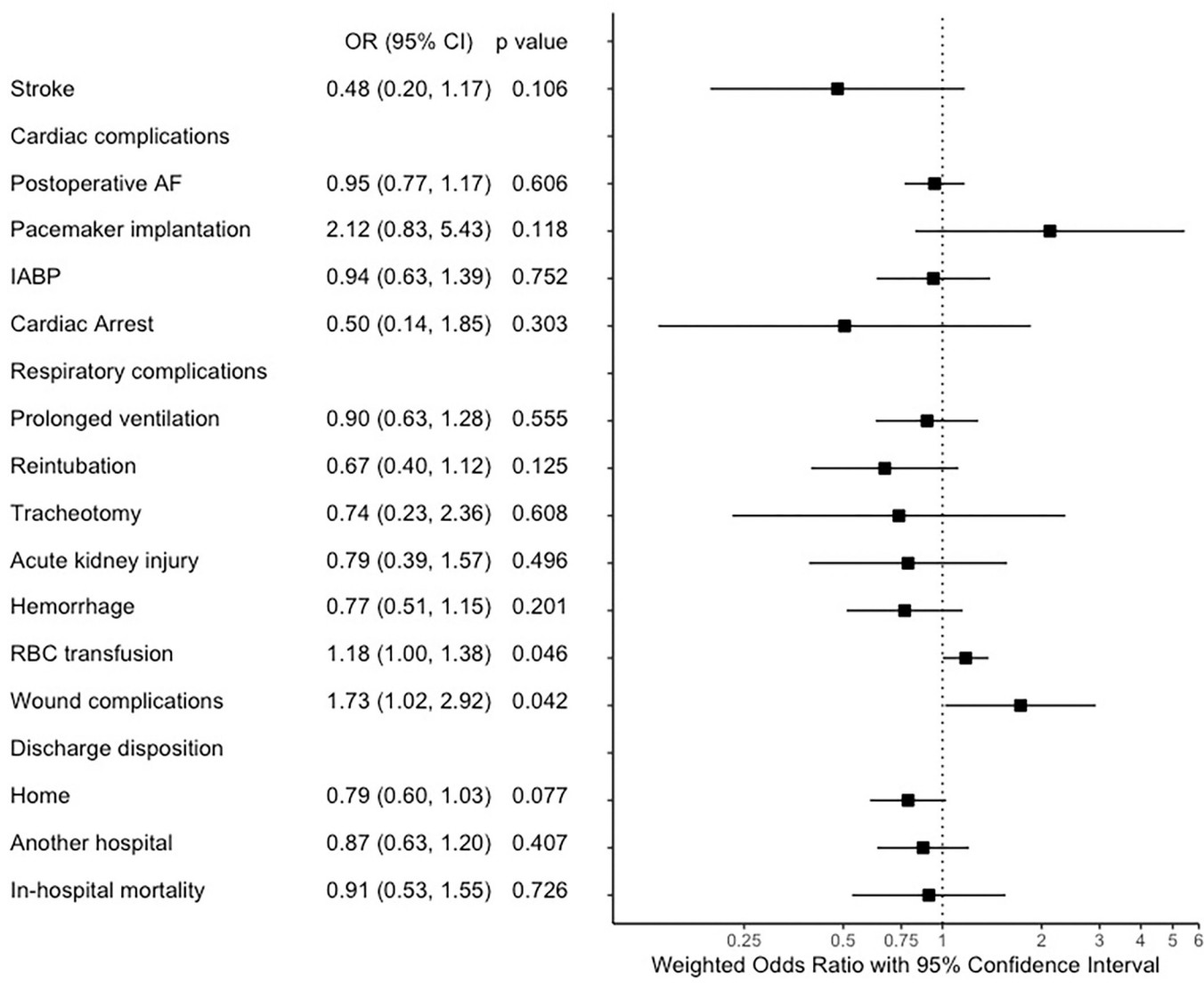

**Fig 4. Odds ratio plot.** Overlap weighted outcome analysis: male vs female.

9.75–16.98) years for the female patient group. The primary endpoint of all-cause mortality at ten years occurred in 1106 patients (26.1%) in the male patient group, compared with 315 (28.5%) in the female patient group. Thirty-day, one, five and ten-year survival rates were 98.8, 96.3, 87.9 and 72.1% in the male patient group and 98.6, 95.6, 88.1, and 69.4% in the female patient group. Fig 5 depicts the unweighted survival function plot for both groups, revealing the worst long-term prognosis for the female cohort (hazard ratio, 1.22; 95% CI, 1.10 to 1.35; $p < 0.001$). Fig 6 illustrates the overlap weighted survival function plot for both groups, where such long-term difference in prognosis disappears (hazard ratio, 0.98; 95% CI, 0.88 to 1.09; $p = 0.693$).

## Discussion

In this contemporary, longitudinal, population-level analysis of patients undergoing primary, isolated CABG, we demonstrated that the female sex is not associated with an increased risk of

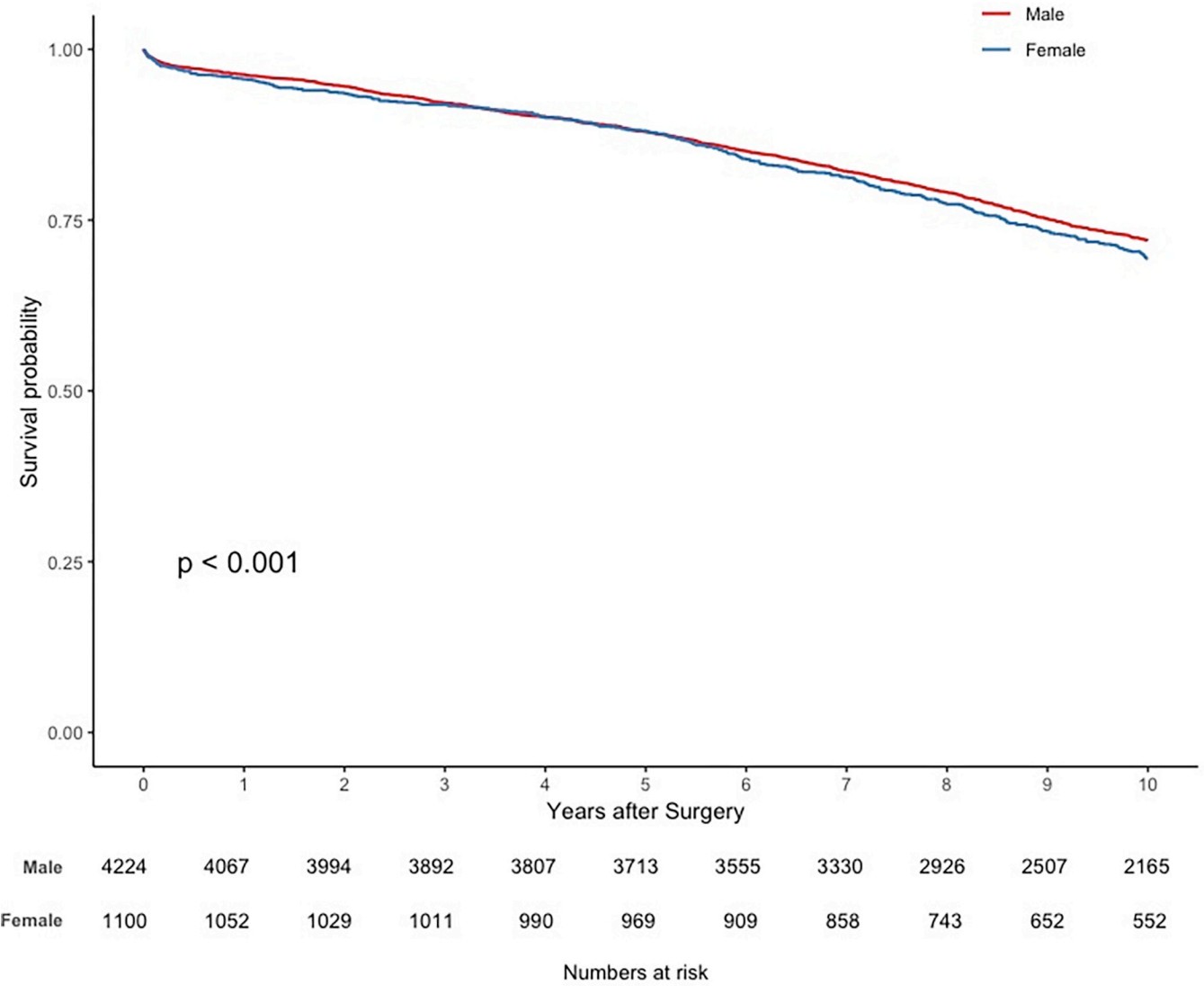

**Fig 5. Unweighted survival function plot.** Unadjusted survival function (Kaplan-Meier method): male vs female.

death at ten-year follow-up, after extensive adjustment for baseline and procedural characteristics.

In our cohort, women were older and had a more substantial burden of comorbidities at baseline. Furthermore, they were less likely to receive additional arterial grafts and received fewer distal anastomosis. Nevertheless, we employed a strategy to control for possible confounding (OW) that emphasizes patients with the most overlap in their observed characteristics [12]. Its corresponding estimand, the average treatment effect in the overlap population, is of natural relevance to this investigation because it highlights the portion of the population where the most treatment equipoise exists in clinical practice. Therefore, the significant difference in 10-year found in the unadjusted analysis faded after down-weighting the extremes of the PS distribution.

In a propensity score-matched analysis of 68774 patients (21.9% women), Guru et al. describe similar survival rates in women to those seen in men at 10-year follow-up [25].

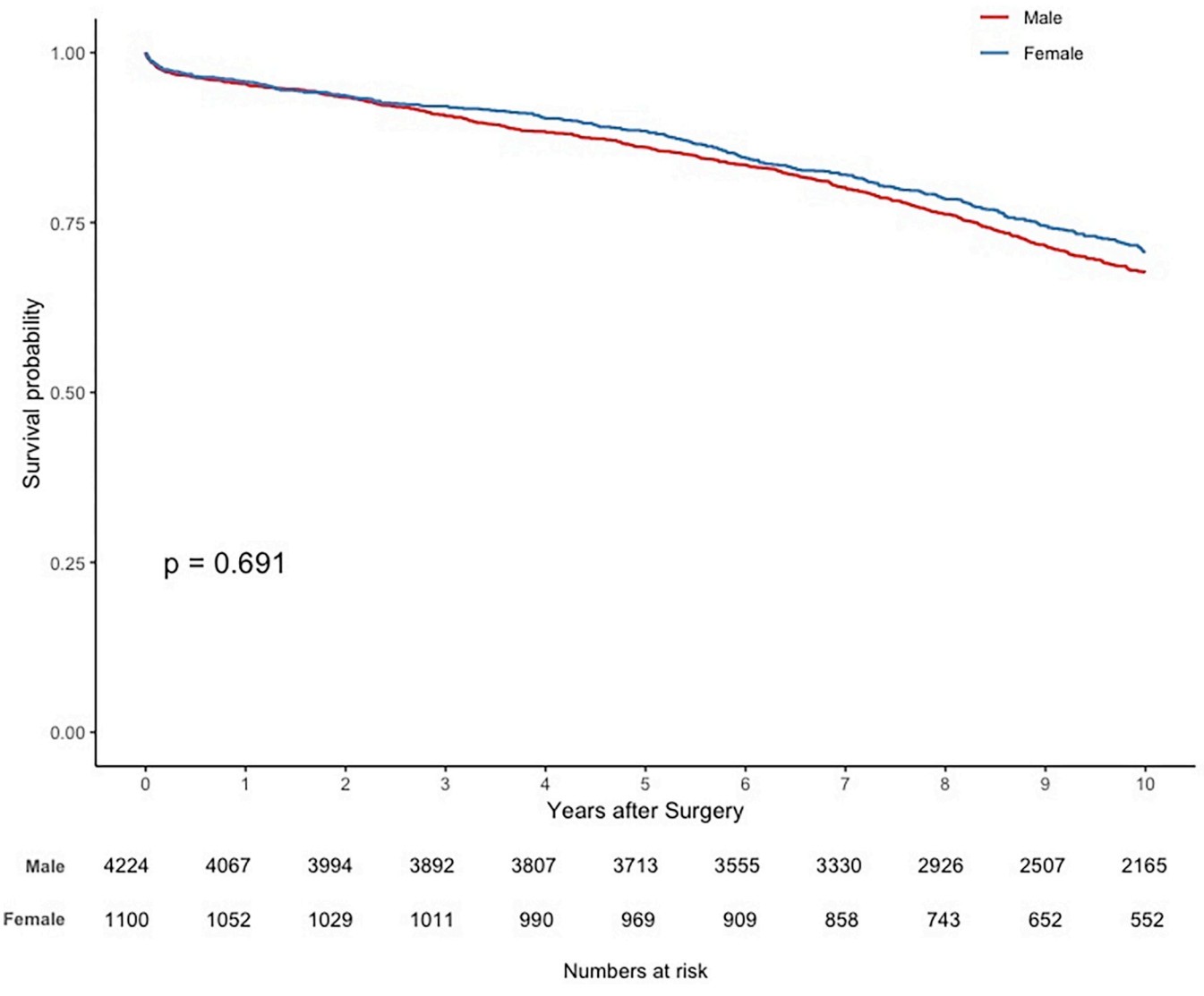

**Fig 6. Overlap weighted survival function plot.** Adjusted survival function: male vs female.

Likewise, Piña et al., from the STICH trial, relate similar all-cause mortality, cardiovascular mortality, and the composite of all-cause mortality or cardiovascular hospitalization, although the small representativeness of women (12%) in the study population [11]. On the other hand, a meta-analysis of 20 observational studies with nearly one million patients (29% women) reported higher mortality in women after CABG at a 5-year follow-up [8]. This result was consistent in the subgroup analysis of prospective and propensity score matching data.

Several observational studies reported higher 30-day mortality rates in women [26–30]. We could not confirm this result based on our data in line with other studies [2, 31–33]. The unadjusted and the overlap weighted cohorts presented similar in-hospital mortality rates. Moreover, our reported in-hospital mortality rates are significantly inferior to those reported in the previously cited meta-analysis [8].

As documented in other series, women represented about 25% of patients undergoing CABG from 2000 to 2015 [25, 33]. Whether this might represent a referral bias is not supported by our data. Women often present atypical symptoms of myocardial infarction, with

attending delays in hospital admission [1]. This should translate in an increased proportion of unscheduled procedures (urgent or emergent), which is contrary to our findings.

There are several limitations to our study. First, although using administrative databases allows for the efficient assessment of large populations over long periods, coding practices were developed for reimbursement issues, not for clinical outcome profiling. As such, imprecise or equivocal definitions may compromise coding accuracy. Additionally, surgical risk models are usually based on a limited number of crucial clinical variables that are typically unavailable in administrative databases [34].

Second, we employed OW to restrict confounding by indication, emphasizing patients with the most overlap in their observed characteristics [12]. Nevertheless, propensity score-based methodologies do not consider factors that are not analyzed, such as patients' frailty, quality of coronary artery targets, quality of venous and arterial conduits, or secondary prevention after CABG. Only a prospective randomized trial, where the distribution of known and unknown confounders would be similar in both the intervention and control groups, could address these issues.

Third, although demonstrating that the gap in CABG outcomes between sexes is narrowing, it would be highly relevant to understand its underlying mechanisms thoroughly.

Our results have important implications for clinical practice, as they might imply a revision of traditional risk scores, which continue to weigh the variable female sex significantly. Furthermore, pursuing an aggressive and timely diagnostic work-up and implementing an adequate revascularization strategy, using multiple arterial grafts and striving for complete revascularization could improve immediate and long-term results in female patients.

## Conclusion

In this longitudinal, population-level analysis of patients undergoing primary, isolated CABG, we demonstrated that the female sex is not associated with increased long-term all-cause mortality compared to their male counterparts. Thus, sex should not influence the undertaking of an adequate revascularization strategy.

## Supporting information

**S1 Table. International Classification of Diseases, 9th Edition, Clinical Modification codes of the conditions defined as exclusion criteria.**
(DOCX)

**S2 Table. International Classification of Diseases, 9th Edition, Clinical Modification codes of the conditions defined as covariates.**
(DOCX)

**S3 Table. International Classification of Diseases, 9th Edition, Clinical Modification codes of the conditions defined as outcomes.**
(DOCX)

**S1 File. Minimal anonymized data set.**
(CSV)

## Acknowledgments

The authors would like to thank doctor Afonso Pedrosa for enabling the linkage of the discharge database to the Portuguese National Patient Registry (RNU) to verify patient life status.

## Author Contributions

**Conceptualization:** Armando Abreu.

**Data curation:** Armando Abreu.

**Formal analysis:** Armando Abreu.

**Methodology:** Armando Abreu.

**Software:** Armando Abreu.

**Writing – original draft:** Armando Abreu.

**Writing – review & editing:** Armando Abreu, José Máximo, Adelino Leite-Moreira.

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
