## [Decision Letter · Decision Letter 0]

1 Aug 2022

PONE-D-22-16893Long-term survival of female versus male patients after coronary artery bypass graftingPLOS ONE

Dear Dr. Abreu,

Thank you for submitting your manuscript to PLOS ONE. After careful consideration, we feel that it has merit but does not fully meet PLOS ONE’s publication criteria as it currently stands. Therefore, we invite you to submit a revised version of the manuscript that addresses the points raised during the review process.

We look forward to receiving your revised manuscript.

Kind regards,

Alessandro Leone, MD

Academic Editor

PLOS ONE

Journal Requirements:

4. Please amend the manuscript submission data (via Edit Submission) to include author Alberto Freitas.

Reviewers' comments:

Reviewer's Responses to Questions

**Comments to the Author**

1. Is the manuscript technically sound, and do the data support the conclusions?

Reviewer #1: Yes

Reviewer #2: Yes

2. Has the statistical analysis been performed appropriately and rigorously? 

Reviewer #1: Yes

Reviewer #2: Yes

3. Have the authors made all data underlying the findings in their manuscript fully available?

Reviewer #1: Yes

Reviewer #2: Yes

4. Is the manuscript presented in an intelligible fashion and written in standard English?

Reviewer #1: Yes

Reviewer #2: Yes

5. Review Comments to the Author

Reviewer #1: This manuscript represents a well written population level analysis investigating female sex and long-term all-cause mortality with revascularization. The data is well described and relatively complete with findings of no prognostic implications with female sex 10 years-procedure. Consistent with national trends an almost 4:1 ratio of men

Reviewer #2: Dear the authors of the manuscript entitled “Long-term survival of female versus male patients after coronary artery bypass grafting”

I want to thank you for your efforts in writing this retrospective study which compares the long-term survival between males and females who underwent coronary artery bypass grafting. This study included good number of patients in both cohorts and had excellent long term follow up period. It is obvious that in the non-adjusted cohorts, that female patients are at having comorbidity risks and received lesser chances of arterial revascularization as well as completeness of revascularization, and that is typical what we believe in and see in our daily practice. Contrary to what we know, and I think that is the beauty of this study, once we match these cohorts, survival was comparable between the 2 cohorts, and that is a fact which is good to provide. Since our decisions in the choice of revascularization strategy are sometimes governed by the gender of the patients, I think this study is good to add to the literature.

My points here to the authors:

1. Did you have any data regarding the syntax score for assessment of the quality of the coronary arteries in both cohorts? I know it is sometimes hard to find this piece of information for all the patients, however if there is available data about this, I guess it will add much to this study

2. Any idea about the echo assessment of the left ventricular function for both cohorts?

3. Do both cohorts had certain type of medication protocol that was identical between the 2 cohorts (Double vs single antiplatelet therapy)

Again, thank you for your efforts in writing this manuscript

6. PLOS authors have the option to publish the peer review history of their article (what does this mean?). If published, this will include your full peer review and any attached files.

Reviewer #1: No

Reviewer #2: **Yes: **salah Eldien Altarabsheh

---

## [Author Response · Author response to Decision Letter 0]

3 Aug 2022

Reply to review

Journal Requirements:

Answer: We have devoted the best of our efforts to meet PLOS ONE’s style requirements. Likewise, files have been renamed as instructed.

Answer: The need for consent was waived by the ethics committee. We have updated the Ethics subsection of our manuscript to include this information.

3. We note that you have indicated that data from this study are available upon request. PLOS only allows data to be available upon request if there are legal or ethical restrictions on sharing data publicly. 

Answer: As requested, we have uploaded the minimal anonymized data set necessary to replicate our study findings as a Supporting Information file. 

4. Please amend the manuscript submission data (via Edit Submission) to include author Alberto Freitas.

Answer: The inclusion of Professor Alberto Freitas as author of the present paper was an unfortunate lapse. Therefore, his name has been removed from the first page of the manuscript.

5. Please include captions for your Supporting Information files at the end of your manuscript, and update any in-text citations to match accordingly. 

Answer: Supporting Information captions have been included at the end of the manuscript in a section titled “Supporting information”. 

Answer: We confirm that our reference list is complete and correct. To the best of our knowledge, none of the cited papers has been retracted. 

Reviewers' comments:

Reviewer's Responses to Questions

Comments to the Author

1. Is the manuscript technically sound, and do the data support the conclusions?

Reviewer #1: Yes

Reviewer #2: Yes

2. Has the statistical analysis been performed appropriately and rigorously? 

Reviewer #1: Yes

Reviewer #2: Yes

3. Have the authors made all data underlying the findings in their manuscript fully available?

The PLOS Data policy requires authors to make all data underlying the findings described in their manuscript fully available without restriction, with rare exception (please refer to the Data Availability Statement in the manuscript PDF file). The data should be provided as part of the manuscript or its supporting information, or deposited to a public repository. For example, in addition to summary statistics, the data points behind means, medians and variance measures should be available. If there are restrictions on publicly sharing data — e.g., participant privacy or use of data from a third party—those must be specified.

Reviewer #1: Yes

Reviewer #2: Yes

4. Is the manuscript presented in an intelligible fashion and written in standard English?

Reviewer #1: Yes

Reviewer #2: Yes

5. Review Comments to the Author

Reviewer #1: This manuscript represents a well written population level analysis investigating female sex and long-term all-cause mortality with revascularization. The data is well described and relatively complete with findings of no prognostic implications with female sex 10 years-procedure. Consistent with national trends an almost 4:1 ratio of men.

Answer: We are most obliged by your kind remarks. 

Reviewer #2: Dear the authors of the manuscript entitled “Long-term survival of female versus male patients after coronary artery bypass grafting”

I want to thank you for your efforts in writing this retrospective study which compares the long-term survival between males and females who underwent coronary artery bypass grafting. This study included good number of patients in both cohorts and had excellent long term follow up period. It is obvious that in the non-adjusted cohorts, that female patients are at having comorbidity risks and received lesser chances of arterial revascularization as well as completeness of revascularization, and that is typical what we believe in and see in our daily practice. Contrary to what we know, and I think that is the beauty of this study, once we match these cohorts, survival was comparable between the 2 cohorts, and that is a fact which is good to provide. Since our decisions in the choice of revascularization strategy are sometimes governed by the gender of the patients, I think this study is good to add to the literature.

My points here to the authors:

1. Did you have any data regarding the syntax score for assessment of the quality of the coronary arteries in both cohorts? I know it is sometimes hard to find this piece of information for all the patients, however if there is available data about this, I guess it will add much to this study.

Answer: We are thankful for your insightful analysis. Unfortunately, the Syntax Score is not routinely calculated in our center. Therefore, this important information is unavailable for analysis.

2. Any idea about the echo assessment of the left ventricular function for both cohorts?

Answer: One of the fundamental problems when using administrative datasets for clinical profiling in cardiac surgery is that certain fundamental variables in all risk score calculations are unavailable. One of such variables is precisely left ventricular function. However, based on the available literature where this item is reported, we should not expect substantial differences in LV function between both groups.

3. Do both cohorts had certain type of medication protocol that was identical between the 2 cohorts (Double vs single antiplatelet therapy)?

Answer: The antiplatelet therapy protocol in use in our department does not make gender-based distinctions. We use single antiplatelet therapy in stable coronary artery disease cases and we use DAPT in all OPCAB cases or CABG following an acute coronary syndrome. Both these variables (clinical presentation and use of CPB) were well balanced at baseline. Therefore, we should not expect differences in antiplatelet therapy between the two groups.

Again, thank you for your efforts in writing this manuscript.

6. PLOS authors have the option to publish the peer review history of their article (what does this mean?). If published, this will include your full peer review and any attached files.

Do you want your identity to be public for this peer review? For information about this choice, including consent withdrawal, please see our Privacy Policy.

Reviewer #1: No

Reviewer #2: Yes: Salah Eldien Altarabsheh

---

## [Decision Letter · Decision Letter 1]

9 Sep 2022

Long-term survival of female versus male patients after coronary artery bypass grafting

PONE-D-22-16893R1

Dear Dr. Abreu

We’re pleased to inform you that your manuscript has been judged scientifically suitable for publication and will be formally accepted for publication once it meets all outstanding technical requirements.

Kind regards,

Alessandro Leone, MD

Academic Editor

PLOS ONE

Additional Editor Comments (optional):

Reviewers' comments:

Reviewer's Responses to Questions

**Comments to the Author**

1. If the authors have adequately addressed your comments raised in a previous round of review and you feel that this manuscript is now acceptable for publication, you may indicate that here to bypass the “Comments to the Author” section, enter your conflict of interest statement in the “Confidential to Editor” section, and submit your "Accept" recommendation.

Reviewer #1: All comments have been addressed

Reviewer #2: All comments have been addressed

2. Is the manuscript technically sound, and do the data support the conclusions?

Reviewer #1: Yes

Reviewer #2: Yes

3. Has the statistical analysis been performed appropriately and rigorously? 

Reviewer #1: Yes

Reviewer #2: Yes

4. Have the authors made all data underlying the findings in their manuscript fully available?

Reviewer #1: Yes

Reviewer #2: Yes

5. Is the manuscript presented in an intelligible fashion and written in standard English?

Reviewer #1: Yes

Reviewer #2: Yes

6. Review Comments to the Author

Reviewer #1: Extensive study created which reviews both historical and large database data regarding females and cardiac surgery. With such large volume of patients and only 16 LTFU this database is powered to reveal trends and the outcome of interest.

Reviewer #2: Dear the authors

Thank you for addressing my reviews comments regarding this manuscript

I have no concerns

Thank you

7. PLOS authors have the option to publish the peer review history of their article (what does this mean?). If published, this will include your full peer review and any attached files.

Reviewer #1: No

Reviewer #2: **Yes: **Salah Eldien Altarabsheh

---

## [Editor Report · Acceptance letter]

15 Sep 2022

PONE-D-22-16893R1 

Long-term survival of female versus male patients after coronary artery bypass grafting 

Dear Dr. Abreu:

I'm pleased to inform you that your manuscript has been deemed suitable for publication in PLOS ONE. Congratulations! Your manuscript is now with our production department. 

Kind regards, 

on behalf of

Dr. Alessandro Leone 

Academic Editor

PLOS ONE